# Effect of Ridging Shapes on the Water–Salt Spatial Distribution of Coastal Saline Soil

**Ji Qi, Kaixiao Sun, Yinghua Pan \*, Qiuli Hu \* and Ying Zhao**

School of Resources and Environmental Engineering, Ludong University, Yantai 264025, China; jiqi0416@outlook.com (J.Q.); M13181116885@126.com (K.S.); yingzhaosoils@gmail.com (Y.Z.)
\* Correspondence: panxingxing@126.com (Y.P.); qiulihu@ldu.edu.cn (Q.H.)

**Abstract:** The Yellow River Delta, located in China, experiences prevalent soil salinization and serves as a crucial ecological management zone within the Yellow River Basin. The shallow groundwater depth and high mineralization contribute to salt accumulation in the soil, which has a negative impact on crop growth. The sustainable use of saline land in the Yellow River Delta hinges on managing the soil salinity within the crop root zone. This study investigated the spatial distribution of soil salinity in coastal saline soil in the Yellow River Delta under various ridging configurations: triangular, arch, and trapezoidal, using flat land as a control. It also examined the impact of evaporation on soil salinity migration. The findings revealed that the ridge–furrow system successfully caused salt to accumulate in the superficial layer of the ridge. Among the three ridge shapes, the triangular ridge was the most effective at concentrating salt on the ridge surface, with 54.04% of the salt mass accumulation in the ridge's top layer (0–1 cm) and with the furrow bottom achieving a maximum desalination rate of 93.07%. The results implied that the triangular ridge fostered a favorable soil environment for crop growth by minimizing the salt content in the furrow. This research provides a theoretical foundation for the sustainable advancement of saline–alkali agriculture in the Yellow River Delta, which can lead to higher crop yields and better land management practices.

**Keywords:** saline soils; micro-landform; water and salt migration; the Yellow River Delta



## 1. Introduction

Soil is an important natural resource for human survival and reproduction, and soil quality directly determines the carrying capacity of land resources, which plays a vital role in human survival and development. Soil salinization has become one of the major problems threatening land resources and food security, and about $7.6 \times 10^7$ hm$^2$ of land worldwide is affected by soil salinization [1]. Therefore, paying attention to the dynamics of salinized soil and taking active countermeasures to prevent the continuous intensification of salinization is an important way to carry out the protective development and utilization of global land reserve resources.

The Yellow River Delta has important ecological value for the global ecosystem, with more than 50% of the total area of salinized soil due to natural conditions [2]. In recent years, due to the shortage of freshwater resources and irrational irrigation, coupled with the shallow groundwater table depth, the salt cannot be discharged for a long time after entering the soil profile, which aggravates soil salinization and seriously restricts the sustainable and high-quality development of the local ecology and agriculture [3–5]. Therefore, appropriate technical measures to control the salt in the soil or accelerate the discharge of salt in the soil profile should be carried out to prevent and control soil salinization.

Overall, it is difficult to improve plant growth or reduce soil salinity in semi-arid areas of China due to water shortages [6]. Ridging is one of the traditional farming methods that improve soil permeability through artificially constructed differences in the ground surface height and has been used for several decades. A previous study suggested that

ridge–furrow plastic mulching brought about a yield increase comparable to irrigated crops but used 24% less water in comparison with irrigation due primarily to a noteworthy water use efficiency and better maintenance of soil water [7]. Studies showed that surface elevations affect spatial soil water and salt distribution, and the soil surface salt content tends to increase with elevation, while water content is the opposite [8–10]. A field study suggested that paired-row planting and furrow irrigation had an increased pod yield, saved water, and enhanced the WUE of groundnut under hot sub-humid conditions, where the ridge and furrow and paired-row methods decreased the ET by 13 and 21%, respectively, and increased the crop WUE by 32.6 and 48%, respectively, over a flatbed [11].

In rain-fed agricultural areas, to solve the problem of irrigation water shortage, furrow sowing is often used, and mulching on the ridge can collect rainfall and provide a suitable water environment for the crops in the trench [12,13]. This method uses the principle of soil water and salt transport to construct a relatively low-salt environment, which has high efficiency, is economically friendly, and has been widely used in practice. A study showed that the concentration of surface soil solution with ridge mulching combined with drip irrigation measures is reduced by 41.25–74.92% compared with flat sowing [14]. Li et al. [15] pointed out that ridge-mulched furrow-sown winter wheat can increase the yield by 25.4% compared with flat land. Sun et al. [16] pointed out that ridge mulching changes the micro-topography of the soil surface, thereby enhancing the water storage capacity of the soil, which can increase the yield by 37–196.4% compared with the control group. Studies showed that in the case of an irrigation water shortage, the yield of furrow-sown winter wheat is significantly higher than that of ridge sowing and has a higher water tolerance threshold, with 7.5 dS/m vs. 5 dS/m, respectively [17]. Trench sowing increased the water content of 0–200 cm soil by 11–15%, and the cereal yield increased by 75% under precipitation conditions of 230–340 mm. Two years of field experiments showed that ridge and furrow mulching technology can significantly improve the water use efficiency and yield of maize [18]. The trench sowing technique effectively reduces soil temperature and prolongs the water supply time during the growing season of winter wheat [19].

These studies showed that soil water and salt are significantly affected by the ridge-furrow planting system. At present, relevant studies mainly study the soil water–salt transport process in the form of ridge cropping from the perspective of the leaching process or the combination of evaporation and leaching, while there are few studies that have considered soil water–salt transport under evaporation alone; when it comes to the geometric characteristic of the ridges, both width and height were studied in the previous research [20–22], though there are few studies involving the influence of ridge shapes on water–salt migration, and the influence of ridge shapes (cross-sectional shapes) on soil water–salt transport and its redistribution pattern is still unclear. Therefore, this study intended to analyze and study the influence of ridges on the spatial distribution of water and salt in the soil profile by constructing different ridge shapes artificially through indoor experiments to explore the regulatory effect of the ridge shape on the spatial distribution of soil water and salt in coastal saline soil to find the suitable ridge shape for creating a low-salinity environment as a supplement to the ridge cropping theory, which has reference significance for the formulation of the "salt avoidance tillage" system in salinized areas.

## 2. Materials and Methods

### 2.1. Soil Sample

The experimental soil was collected from the Base of Modern Agriculture High-Quality Development of Modern Agriculture in Dongying (37.66′ N, 118.92′ E), and the depth of the soil was 0–60 cm. After the soil sample was dried, it was sieved (2 mm) for reserve, and its physical and chemical properties are shown in Table 1.

**Table 1.** Basic physical and chemical properties of the experimental soil samples.

| Indicators | Soil Particle Size Composition (%) | | | Nitrogen (g/kg) | Salt Content (g/kg) | pH | Soil Texture |
|---|---|---|---|---|---|---|---|
| | Sand | Silt | Clay | | | | |
| Measured values | 81.73 | 10.76 | 7.51 | 0.47 | 9.56 | 7.9 | Sandy loam |

### 2.2. Experimental Setup

The experiments were carried out in the Laboratory of Regional Soil and Water Resources and Agricultural Utilization of Ludong University, and the designed shapes of the ridge profiles were triangular, arched, and trapezoidal, denoted as T1, T2, and T3, respectively, using flat surface (CK) as the control. The experimental setup consisted of transparent cylindrical plexiglas tanks (8 mm thick) with a size of $60 \times 10 \times 30 \text{ cm}^3$ and Mariotte bottles. A water inlet pipe and water level control pipe were provided in the lower part of the clay tank to create a groundwater environment and control the groundwater table depth. To create uniform groundwater conditions and ensure good water permeability, 2 cm thick quartz sand was poured into the bottom of the soil tank and covered with clean and pollution-free filter paper to separate the quartz sand from the experimental soil, where the bulk density of the loaded soil was $1.35 \text{ g/cm}^3$. After reaching 13 cm, an artificial ridge treatment was constructed using subsequent soil, and the maximum height of the three ridge shapes was about 15 cm.

After the soil tank was filled, under the condition that the Mariotte bottle did not leak and the distilled water was sufficient, the Mariotte bottle was connected to the water supply pipe at the bottom of the organic glass soil tank with a rubber hose to maintain a fixed water head to form a fixed water supply condition. When the water level of the soil tank reached the highest point of the ridge, the water supply was stopped, and the soil tank was placed indoors without direct sunlight to create a natural evaporation condition. The experimental soil samples were collected 90 days after the end of the water supply, and soil samples were taken every 2 cm of soil with a depth of 1 cm along the surface to study the distribution of salt along the ridge surface, where the sampling volume was $2 \times 10 \times 1 \text{ cm}^3$. The sampling volume of the $2 < d \leq 16$ cm depth profile soil was $5 \times 10 \times 2 \text{ cm}^3$. Samples were collected with a sharp soil dressing knife to a given size, where the center coordinate of each soil sample was recorded as the sample coordinate. The sample point coordinates were drawn in a two-dimensional Cartesian coordinate system and are shown in Figure 1. The soil water content was determined immediately after sampling, and the rest of the samples were placed in a ziplock bag after drying, grinding, and passing through a 2 mm sieve for testing.

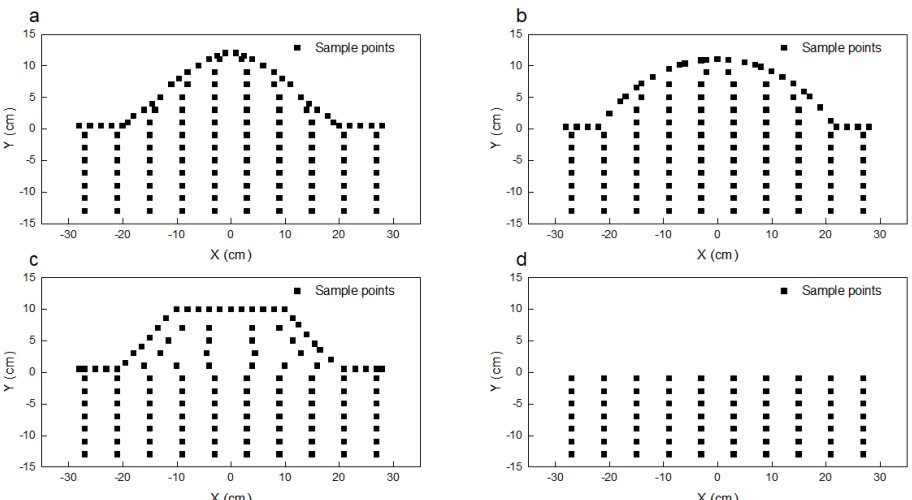

**Figure 1.** Distribution of the sample points; (**a**–**d**) represents T1, T2, T3, and CK, respectively.

### 2.3. Test Items

The moisture content of the soil samples (105 °C, 8 h) was determined using the oven drying method. The conductivity of the soil sample leachate (5:1 water–soil ratio) was determined using a DDS-12A conductivity meter. The Cl$^-$ content in the soil samples was determined using the silver nitrate titration method. The Na$^+$ content in the soil samples was determined using the flame photometer method, where the operating procedure was based on *Soil Agricultural Chemistry Analysis* [23].

### 2.4. Data Analysis

The desalination rate ($D$) is an important indicator used to characterize soil salinity changes, and the formula is

$$D = (T - E)/T \tag{1}$$

where $D$ is the desalination rate and $T$ is the initial soil salinity content, $E$ is the existing salt content of the soil, $D > 0$ indicates the soil desalination, and $D < 0$ indicates the soil salt accumulation.

The coefficient of variation, which reflects the degree of spatial variation of soil physical and chemical indexes, is calculated using the following equation:

$$C_v = SD/\bar{X} \tag{2}$$

where $SD$ is the standard deviation of the sample and $\bar{X}$ is the mean of the sample. $C_v < 10\%$ indicates a weak variation, $10\% \leq C_v \leq 100\%$ indicates a moderate variation, and $C_v > 100\%$ is a strong variation [3].

## 3. Results

### 3.1. Spatial Distribution of the Soil Water Content in Different Ridge Shapes

The soil water content is an important carrier of salt, where in the process of water absorption and infiltration, water carries salt in the soil profile, and after the water reaches the surface, it is lost due to evaporation to form water vapor and the salt is retained in the soil.

#### 3.1.1. Soil Water Content (0–1 cm)

Figure 2 shows the water content of the topsoil with different ridge treatments at the end of the experiment. The differences in surface soil water content between the ridges and furrows can be seen. The water content of the furrow surface of the three ridge forms was higher than that of the ridge surface, the water content showed a decreasing trend from the furrow to the ridge, and the water content was lower at the convex or inflection point of the ridge, where these latter locations also had the most salt accumulation. According to the experimental data, the average moisture content of the surface layer at the furrow of T1 was 14.09%, the average moisture content of the ridge surface layer was 12.33%, the average moisture content of the surface layer of T2 furrow was 14.66%, the average moisture content of the ridge surface layer was 12.45%, the average moisture content of the surface layer of T3 furrow was 13.21%, the average moisture content of the ridge surface layer was 12.42%, and the average moisture content of CK surface layer was 12.85%. The average water content of the surface layer of the three ridge forms was higher than that of the flat surface, which can provide better water conditions for plant growth and salt leaching.

It can be seen from Figure 2 and Table 2 that the surface 0–1 cm soil water content fluctuated after ridging, and the average soil water content of the surface layer (0–1 cm) of the furrow was higher than that of the soil water content at the ridge surface layer (0–1 cm), and the surface soil water content showed a trend of reduction to some extent during the transition from a furrow to a ridge in each treatment. After evaporation, the surface water contents of the T1, T2, and T3 ridges were 4.07%, 3.18%, and 13.18% lower than that of the CK surface, respectively, and the average water content of the T3 ridge was 11.16%, which

was lower than those of T1, T2, and CK with 12.33%, 12.45%, and 12.85%, indicating that the surface of the trapezoidal-shaped ridge had greater evaporation strength. The surface water contents of T1, T2, and T3 were higher than that of CK, and were 9.65%, 14.03%, and 2.76% higher, respectively.

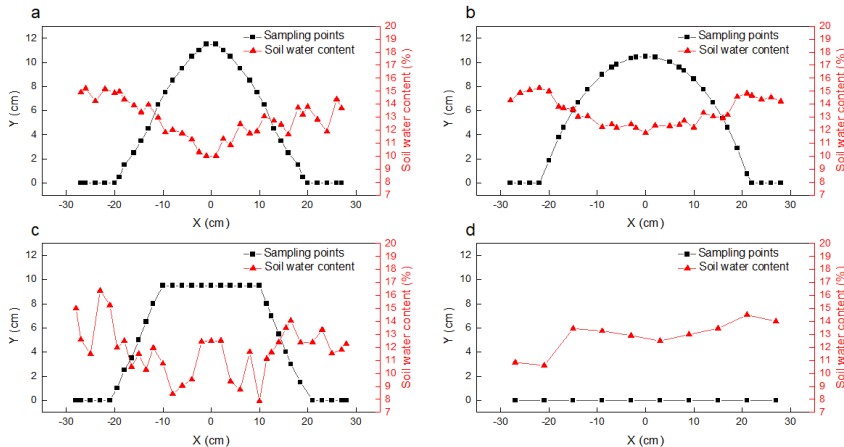

**Figure 2.** Variation in soil water content in the superficial layer (0–1 cm) under different treatments; (**a–d**) represents T1, T2, T3, and CK, respectively.

**Table 2.** Statistical results of soil indicators of 0–1 cm soil in the lower layer of different treatments.

| Treatments | Parameters | Positions | Maximum | Minimum | Average | STD | Coefficient of Variation (%) |
|---|---|---|---|---|---|---|---|
| T1 | Water content (%) | Ridge | 14.97 | 10.02 | 12.33 | 1.35 | 10.93 |
| | | Furrow | 15.2 | 11.89 | 14.09 | 1.08 | 7.66 |
| | Salinity (g/kg) | Ridge | 135.94 | 5.62 | 48.78 | 38.66 | 79.25 |
| | | Furrow | 16.83 | 7.26 | 12.92 | 2.83 | 21.86 |
| T2 | Water content (%) | Ridge | 15 | 3.09 | 12.45 | 2.31 | 18.58 |
| | | Furrow | 15.25 | 14.22 | 14.66 | 0.38 | 2.59 |
| | Salinity (g/kg) | Ridge | 109.35 | 26.61 | 72.68 | 26.87 | 36.98 |
| | | Furrow | 42.34 | 22.65 | 33.79 | 6.51 | 19.25 |
| T3 | Water content (%) | Ridge | 14.05 | 7.85 | 11.16 | 1.68 | 15.01 |
| | | Furrow | 16.36 | 11.5 | 13.21 | 1.73 | 13.09 |
| | Salinity (g/kg) | Ridge | 179.64 | 42.13 | 92.68 | 32.23 | 34.77 |
| | | Furrow | 107.19 | 37.92 | 53.56 | 19.98 | 34.31 |
| CK | Water content (%) | | 14.5 | 10.61 | 12.85 | 1.25 | 9.76 |
| | Salinity (g/kg) | | 46.86 | 31.6 | 40.74 | 4.22 | 6.68 |

Note: T1, T2, T3, and CK represent triangular, arched, trapezoidal, and flat, respectively.

By calculating the coefficient of variation of the ridge and furrow surface of different treatments, the influence of ridging and the shapes of ridges on the fluctuation of soil surface water content were analyzed (Table 2). It can be seen from Table 2 that after the evaporation, the water content of the surface layers of the T1, T2, and T3 ridges all showed medium variations, and the coefficients of variation were 10.93%, 18.58%, and 15.01%, respectively, which were higher than the coefficient of variation of 9.76% in the surface layer of CK; furthermore, T2 was the largest, indicating that the ridge changed the redistribution of soil surface water after evaporation, and the arch-shaped ridge led to more obvious water fluctuations.

### 3.1.2. Distribution of the Water Content in the Soil Profiles

Figure 3 shows the soil water content of the ridge and furrow soil profile after evaporation, and Figure 4 shows the contour diagram of the soil profile water distribution after evaporation. It can be seen from Figures 3 and 4 that the surface water content of each treatment was the smallest, and with the increase in depth, the soil water content showed an increasing trend and then tended to be flatter. The water contents of T3 and CK increased first and then decreased with the increase in depth, and the highest water content of CK was 15.01% at a depth of 5 cm. The highest water content of the T3 ridge was 16.79% at a depth of 13 cm, and the maximum water content at a depth of 5 cm in the furrow profile was 16.45%. It can be seen from Figure 3a that the soil water content increased below 0–1 cm in the ridge surface layer, and the soil water content of the top layer of different treatments showed a trend of T1 < T3 < T2 < CK, i.e., 10.02%, 10.26%, 12.10%, and 12.85%, respectively.

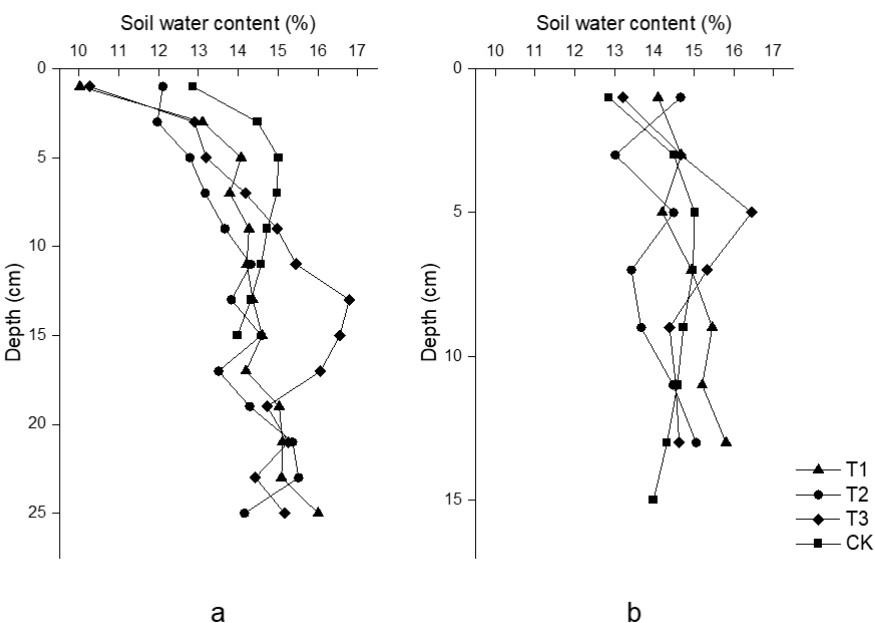

**Figure 3.** Trends of the soil water content in different ridge and furrow profiles, where (**a**,**b**) represent ridge and furrow profiles, respectively.

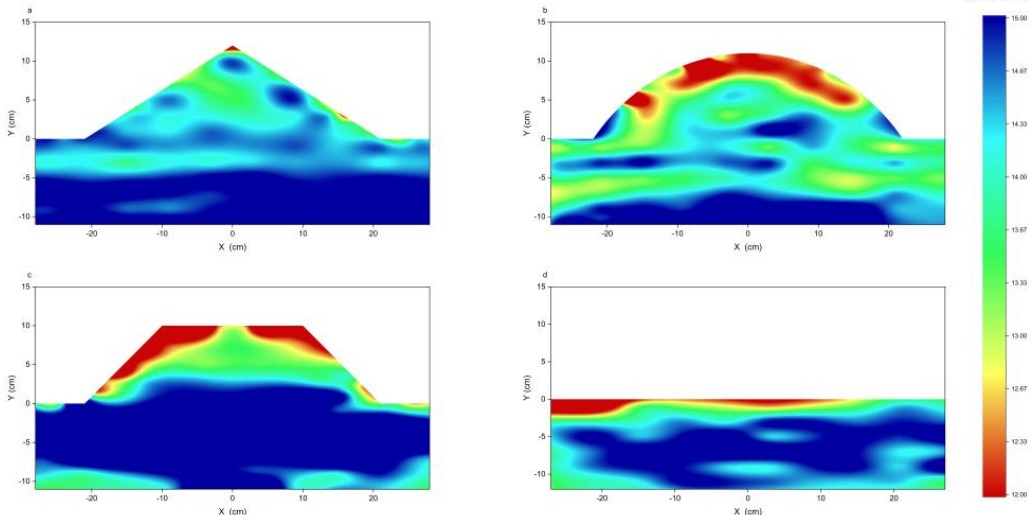

**Figure 4.** Spatial distribution of water content in the soil profile under different treatments, where (**a**–**d**) represents T1, T2, T3, and CK, respectively.

In contrast, the water content of the subsoil was opposite to that of the surface layer, and the soil water content was ranked CK < T2 < T3 < T1, i.e., 13.98%, 14.16%, 15.16%, and 16.01%, respectively. At a depth of 5 cm, the water content of ridge soil was ranked T2 < T3 < T1 < CK, i.e., 12.78%, 13.20%, 14.07%, and 15.01%, respectively. Within the depths of 0–10 cm, the water contents of T1, T2, and T3 were higher than that of CK, and gradually became consistent below the 10 cm soil layer, indicating that the ridge part had a greater evaporation strength. As shown in Figure 3b, the water content of the soil profile of the furrow increased with the increase in depth. Overall, the water content of the furrow was higher than that of the ridge, indicating that the ridge was conducive to maintaining the soil water. The vertical coefficient of variation of the ridge water content was 7.95–11.90%, while that of the furrow was 4.2–6.67% compared with the 4.86% of CK.

### 3.2. Spatial Distribution of the Soil Salinity in Different Planting Ridge Shapes

Creating a low-salinity growth environment for crops in areas affected by salinization is of critical importance. The main purpose of this study was to use the ridge furrow treatment to create salt accumulation in some parts of the soil profile and a desalination situation in other parts to form a low-salt area to facilitate plant growth and development. Furthermore, for the salt in the high-concentration area, salt removal can be carried out artificially to reduce the overall salt content of the profile.

3.2.1. Spatial Distribution of the 0–1 cm Soil Salinity

After the experiment, the salt in the soil was redistributed, where 60.01%, 40.46%, 47.81%, and 38.62% of the total salt mass were concentrated in the soil surface layer of T1, T2, T3, and CK in 0–1 cm, respectively. The accumulation ratio of the surface salt in the T1 ridge was the largest at 54.04%, while the total surface salt content under the same projection area of soil without ridge accounted for only 23.46% of all salt in the soil tank, which was much lower than that of other treatments. Figure 5 shows the trend of 0–1 cm soil salinity in the surface layer after evaporation. It can be seen from the figure that the surface soil was in a state of salt accumulation; from the furrow to the ridge, the salt content of the surface soil showed the opposite trend to the water content, that is, the salt content of the surface soil gradually increased, and especially the changing trend in salt content on the triangular-shaped ridge was the most obvious. After a statistical analysis of the soil water and salt content, the results are presented in Table 2. It can be seen that the average salt content of the ridge surface soil was ranked T3 > T2 > T1, i.e., 92.68, 72.68, and 48.78 g/kg, respectively, and the salt content of furrow surface soil was ranked the same with 53.56, 33.79, and 12.92 g/kg, respectively. The average salt content of the surface layer of the T3 ridge was the highest, and the difference between the ridge and furrow was the largest, indicating that the trapezoidal ridge had the most obvious effect on the salt aggregation, among which the average salt contents of the T1 and T2 furrows were lower than that of CK (40.74 g/kg), and T1 was the lowest. After the construction of the ridge furrow system, a relatively low-salt environment could be created for the furrow to a certain extent, among which from the current situation, the effect of the triangular ridge is better.

From the experimental results, ridge raising made the salt change of the ridge more drastic, which was reflected in the extremely high soil salt content at the geometric inflection point of the ridge. The analysis of the coefficient of variation of soil salinity content in the surface layer of ridges and furrows showed that the spatial variability of soil salinity after evaporation was moderate on the surface. The maximum coefficient of variation of the surface layer of the T1 ridge was 79.25%, which was higher than that of the 36.98% of T2 and 34.77% of T3, indicating that the salt content of the triangular ridge layer fluctuated the most. The lowest position of the ridge salt content in Figure 5b was due to an obvious crack during the drying process, resulting in sudden changes in the soil salt content. The trapezoidal ridge (Figure 5c) also showed an abnormal salt content measurement result due to cracks at the right inflection point.

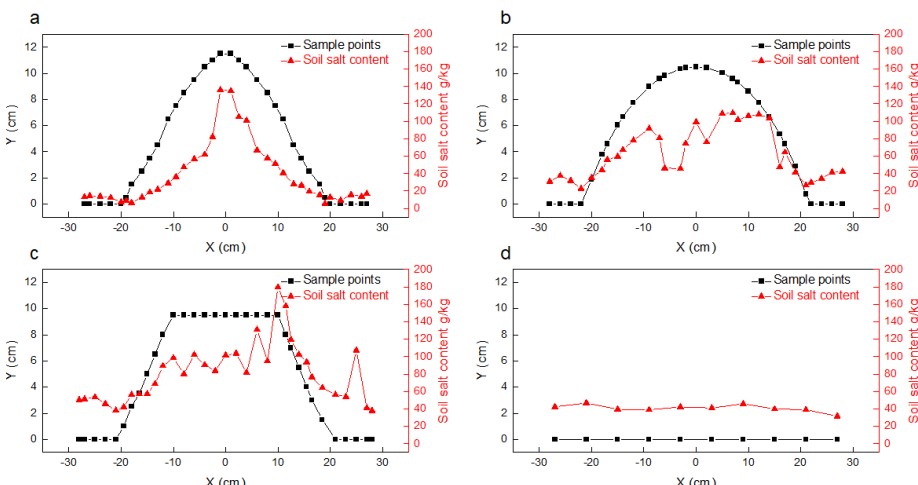

**Figure 5.** Variation in the soil salinity in the superficial layer (0–1 cm) under different treatments, where (**a–d**) represents T1, T2, T3, and CK, respectively.

3.2.2. Spatial Distribution of the Salinity in the Soil Profiles

Figures 6 and 7 show the vertical variation and spatial distribution of the salt in the soil profiles under different treatments, and it can be seen from the figures that the soil salinity accumulated on the surface layer of the ridge after evaporation. The proportion of salt in the ridge to the total salt content was ranked T1 > T2 > T3, i.e., 81.77%, 80.26% and 73.11%, respectively, and the salt aggregation effect was the best with the T1 treatment. As shown in Figure 6a, in the range of 0–5 cm, the salt content of each treatment decreased rapidly with increased depth, where T1, T2, T3, and CK decreased by 124.6, 63.3, 80.35 and 29.51 g/kg, respectively, and the soil salt content was ranked T2 > T3 > CK > T1 at a depth of 5 cm, indicating that the salt content of the triangular ridge decreased fast with depth. Below the depth of 5 cm, the soil salt content under the T1, T2, and T3 ridges continued decreasing, and the final bottom salt content was ranked T2 > T3 > T1, i.e., 2.18, 0.81, and 0.50 g/kg, respectively. The salt content of CK remained basically unchanged until the bottom salt content, which was 9.44 g/kg. The ridges of T1, T2, and T3 showed desalination from depths of 7, 9, and 11 cm, respectively, and the desalination rates were 60.85%, 89.22%, and 58.90%, respectively. With the increase in depth, the desalination rate also increased gradually, and the maximum desalination rates reached 94.76%, 93.82%, and 91.98%, respectively, and the triangular-shaped ridge system had the best desalination efficiency.

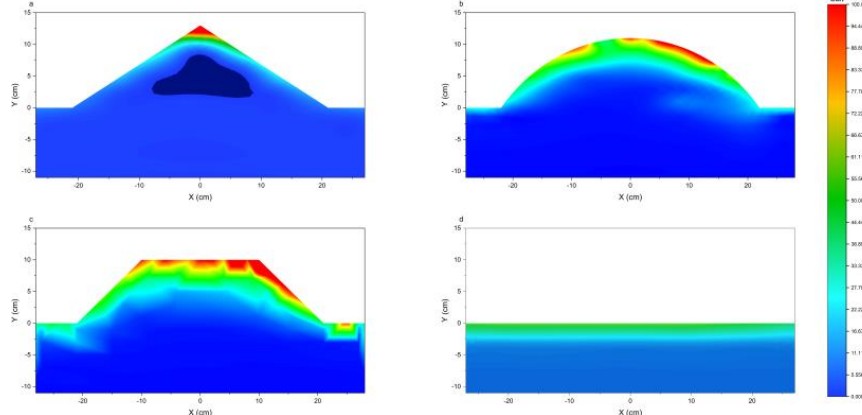

**Figure 6.** Spatial salt distribution in the soil profile with different treatments, where (**a–d**) represents T1, T2, T3, and CK, respectively.

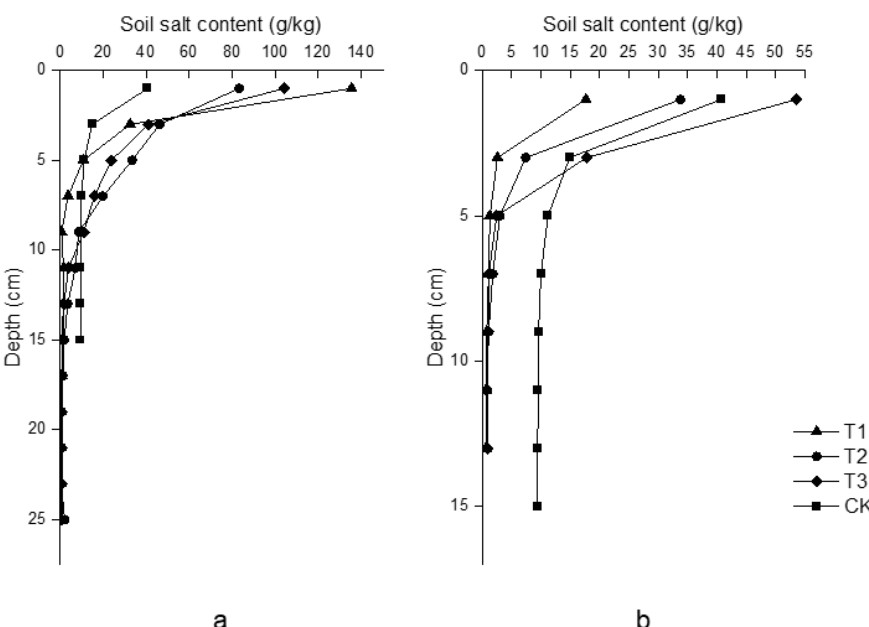

**Figure 7.** Trends of the soil salinity in different ridge and furrow profiles, where (**a**,**b**) are ridge and furrow profiles, respectively.

As shown in Figure 6b, the changes in soil salinity of furrows under different ridge-furrow systems were similar to those of the ridges, and the soil salinity decreased rapidly in the range of the 0–5 cm soil layer, where T1, T2, and T3 decreased by 16.36%, 30.29%, and 6.74%, respectively. At the depth of 3 cm, the soil desalination rates were ranked T1 > T2 > CK > T3, i.e., 72.69%, 22.38%, −56.22%, and −86.54%, respectively. At the depth of 5 cm, the salinity of the T3 furrow decreased rapidly, and the soil desalination rates were ranked T1 > T3 > T2 > CK, i.e., 86.30%, 74.10%, 68.09%, and −17.50%, respectively. The desalination rate did not change much below 5 cm, where T1 maintained a high desalination rate in each treatment, and the maximum desalination rates of T1 and T2 appeared at the bottom (15 cm), which were 93.07% and 91.63%, respectively. On top of that, T3 and CK had desalination rates of 92.01% and 0.96% in the 13 cm deep soil layer, respectively, indicating that the ridge–furrow patterns were conducive to desalination, among which the triangular ridge had the largest desalination rate, indicating that the desalination efficiency of this type of ridge–furrow system was the best, followed by the trapezoidal ridge, and the round arch ridge was the worst.

Na$^+$ and Cl$^-$ are the key ions that hinder crop growth. It can be seen from the data that Na$^+$ and Cl$^-$ were concentrated at the ridge, and their distribution patterns in the surface layer were similar to the salt content. The average contents of Na$^+$ in the T1, T2, and T3 ridges were higher than that on the surface of CK, and the average content of Na$^+$ in the T3 ridges was the highest at 9.25 g/kg. The average soil Na$^+$ content at the furrows was ranked T2 > T3 > CK > T1, indicating that the average content of Na$^+$ in the surface layer of the T1 furrow was the lowest in each treatment and was 1.46 g/kg. The average Cl$^-$ contents of the soil in the CK and T1, T2, and T3 ridges were 42.40, 38.83, 90.01, and 46.46 cmol/kg, respectively, among which T2 was the highest, indicating that the trapezoidal-shaped ridge had the best effect on Cl$^-$ aggregation. The average Cl$^-$ contents in the surface soils of T1, T2, and T3 were 10.58, 30.48, and 43.87 cmol/kg, respectively, of which T1 was the lowest, indicating that the triangular-shaped ridge was more likely to create an environment with lower harmful ion levels in the furrows.

## 4. Discussion

Soil salinization occurs due to the movement of water through a porous material, which brings salt to the surface. As the water evaporates, the salts accumulate on the exterior surfaces or within the top layer of soil [24–26]. Theoretically, reducing water evap-

oration can reduce the degree of soil salinization and improve the growing environment of crops. Furrow planting is commonly used in agricultural production and is widely used around the world. In this study, after ridging, due to its lower positions, the airflow speed on the furrow surface was slower, and thus, the evaporation intensity was low, resulting in water gathering in the ridge and evaporation loss through the ridge surface, and the water content at the ridge and furrow was improved compared with the flat treatment, which was consistent with the results of Sun Chitao et al. [14]. The average annual evaporation in the Yellow River Delta is about four times the average annual precipitation, 70% of the precipitation occurs in summer, and has a shallow groundwater table depth, resulting in upward accumulation of soil salinity in spring and autumn rather than downward leaching [27,28]. After ridge lifting, the salt is transported to the ridge due to its strong evaporation strength, which reduces the soil salinity at the furrow [29,30]. Naturally, the salt at the surface of the furrow is dissolved and transported downward due to the heavy precipitation in summer, which further reduces the soil salt content at the ridge and furrow [31]. The soil surface evaporation was uniform under flat cropping, and there was no significant difference in the soil surface water content and salt content.

This study found that the migration of water and salt in the soil profile was also affected by the ridge shape, and the low soil water content and high salinity often appeared at the geometric inflection point of the ridge top, which may be caused by a large airflow velocity and high evaporation intensity at this position [29,30]. Taking the trapezoidal-shaped ridge as an example, our hypothesis was that in the process of water salt migrating up to the ridge, when passing through the geometric inflection point of "convexity", a part of the water first evaporated and left the salt here; due to the effect of the matrix potential and solute potential, water salt was more easily caught at this point. Furthermore, different from the geometric inflection point on the ridge, the geometric inflection point of the "inner concave" of the ridge and furrow had no obvious salt accumulation due to the small evaporation intensity and limited water and salt reserves in the lower part. The surface layer of the triangular ridge had the highest salinity, which may have been due to the influence of the resultant force when the soil with salt was moved up; therefore, in the process of salt evaporation migration, there was both matrix potential traction from above and matrix potential traction from the soil surface (microtopographic edge), resulting in the gradual accumulation of surface salt all the way up through the slope, and the highest salt concentration happened at the superficial layer of the top, which also indicated that the strength of salt accumulation was related to the elevation of the point [9]. Through our study, 47.81%~60.01% of the total salt mass of each treatment was found concentrated at the superficial layer of the soil, which was higher than in the CK (38.62%), indicating a better salt accumulation effect on the ridge. The results provided observational evidence of salt migration through capillary rise [32].

The high concentration of $Na^+$ outside the cells of saline–alkali plants and the negative membrane potential in the plasma membrane constitute a transmembrane electrochemical potential difference, which promotes the excessive accumulation of $Na^+$ in the cell, causing an imbalance of intracellular ion metabolism and ion toxicity [33]. After ridge lifting, the soil $Na^+$ showed a similar distribution pattern with salt, when the ridge shape was triangular, the ridge and furrow soil $Na^+$ was the lowest, as well as the salt content, indicating that the effect of the triangular ridge was better for both salt and $Na^+$ removal, which may have been because of the flat and unobstructed triangular ridge slope, which made it possible to gather it on the surface. Meanwhile, due to the limited size of indoor experimental devices, the effect of the ridge shape on soil water and salt transport in fields still needs to be further explored.

## 5. Conclusions

Following the ridging creation, the soil's micro-topography resulted in variations in both soil water and salt distribution. The distribution of water and salt in different ridge patterns can serve as a guide to understanding capillary rise, which implies a potential

mechanism for the upward movement of salt in the ridge–furrow planting pattern. The ridge and furrow surface layer had a higher water content than the ridge, while salt was concentrated on the ridge, with the highest salt content at the geometric inflection point. The overall salinity in the ridge and furrow areas decreased, creating favorable conditions for plant growth. Notably, the soil within the 0–1 cm range of the triangular ridge's surface layer accumulated 54.01% of the entire soil profile's salt content. The average content of $Na^+$ and $Cl^-$ of the surface layer of the T1 furrow was the lowest. Furthermore, the maximum desalination rate at the furrow bottom was the highest, reaching 93.07%, suggesting that the triangular ridge's desalination efficiency outperformed other treatments. This study offers a scientific basis for the continual improvement of saline–alkali farming in the Yellow River Delta, a progression that can result in increased crop production and more effective land stewardship techniques.

**Author Contributions:** Conceptualization, Y.P.; methodology, Y.P.; validation, Q.H. and Y.P.; formal analysis, J.Q. and K.S.; investigation, J.Q.; resources, Y.P.; data curation, J.Q. and Y.P.; writing—original draft preparation, J.Q.; writing—review and editing, J.Q., K.S., Y.P., Y.Z. and Q.H.; visualization, J.Q. and K.S.; supervision, Y.P.; project administration, Y.P.; funding acquisition, Y.P. All authors have read and agreed to the published version of the manuscript.

**Funding:** This research was funded by the Taishan Scholars Youth Expert Program, China (201812096); the National Natural Foundation of Shandong Province (ZR2021MD036); and the "Youth Entrepreneurship Science and Technology Program" of Shandong Provincial Colleges and Universities (2019KJF017).

**Data Availability Statement:** All data reported here are available from the authors upon request.

**Conflicts of Interest:** The authors declare no conflict of interest.

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
