# Peer review of "Effect of Ridging Shapes on the Water–Salt Spatial Distribution of Coastal Saline Soil"

_water, doi:10.3390/w15162999_

Round 1

Reviewer 1 Report (Previous Reviewer 4)

The manuscript has been substantially improved

Author Response

We appreciate your letter and the insightful feedback from the reviewers regarding our manuscript. This constructive critique was invaluable and significantly beneficial. We have meticulously reviewed these comments and made the necessary amendments. All changes made to the text are highlighted.

Reviewer 2 Report (Previous Reviewer 3)

The current manuscript is much better. I accept it for publication.

Author Response

We appreciate your letter and the insightful feedback from the reviewers regarding our manuscript. This constructive critique was invaluable and significantly beneficial. We have meticulously reviewed these comments and made the necessary amendments. All changes made to the text are highlighted.

Reviewer 3 Report (Previous Reviewer 1)

General comment:

The paper provides a clear overview of the study, focusing on the spatial distribution of soil salinity in coastal saline soil in the Yellow River Delta. It highlights the significance of managing soil salinity for sustainable land use and crop growth. The abstract effectively summarizes the research findings and their implications for the advancement of saline-alkali agriculture in the region.

Specific comments:

  1. The paper could benefit from a brief explanation of the methodology employed in the study.
  2. It would be helpful to mention the sample size or area covered in the research.
  3. The document could provide more context regarding the specific crops or vegetation considered in the study.
  4. Including information on the time period and duration of data collection would add further clarity.
  5. The paper could mention the significance of the findings in the broader context of agricultural practices in coastal saline areas.

Questions:

  1. What are the key factors contributing to soil salinity in the Yellow River Delta?
  2. How did the researchers measure and analyze soil salinity in the different ridge configurations?
  3. What were the specific findings regarding the impact of evaporation on soil salinity migration?
  4. How do the results of this study contribute to the existing knowledge on managing soil salinity in coastal saline areas?
  5. Are there any limitations or areas for further research identified in the abstract?

Constructive feedback:

This article focuses on the spatial distribution of soil salinity in the Yellow River Delta, China. It investigates different ridge configurations and their impact on salt accumulation in coastal saline soil. The findings demonstrate the effectiveness of the triangular ridge shape in minimizing salt content in the furrow and fostering a favorable soil environment for crop growth. The paper provides valuable information on the spatial distribution of soil salinity and the effectiveness of different ridge configurations in managing salt accumulation. To enhance the text, it would be beneficial to address the specific methodology used, include more contextual details, and highlight the broader implications of the research. Additionally, mentioning limitations or areas for further study would provide a more comprehensive perspective.

  1. The paper lacks specific details regarding the methodology employed in the study. The readers would benefit from knowing how soil salinity was measured, the sample size or area covered, and the statistical analysis techniques used. Without this information, it is challenging to assess the scientific rigor of the research.

  2. While the abstract mentions the negative impact of soil salinity on crop growth, it does not provide concrete evidence or quantitative data to support this claim. Including specific findings or statistics on crop yield reduction or growth inhibition due to soil salinity would strengthen the argument.

  3. The text briefly mentions the impact of evaporation on soil salinity migration but does not delve into the details. It would be helpful to provide more information on the specific mechanisms or processes involved and how they contribute to the overall findings.

  4. Although the abstract discusses the implications of the research for sustainable agriculture in the Yellow River Delta, it does not address potential limitations or challenges in implementing the triangular ridge configuration on a larger scale. It would be beneficial to acknowledge any constraints or feasibility issues that may arise when applying these findings in practical farming scenarios.

Summary:

The research offers a theoretical foundation for sustainable saline-alkali agriculture in the region, aiming to improve crop yields and land management practices.

Author Response

We appreciate your letter and the insightful feedback from the reviewers regarding our manuscript. This constructive critique was invaluable and significantly beneficial. We have meticulously reviewed these comments and made the necessary amendments. All changes made to the text are highlighted.

Reviewer 4 Report (Previous Reviewer 5)

 It is accepted

No comment 

Author Response

We appreciate your letter and the insightful feedback from the reviewers regarding our manuscript. This constructive critique was invaluable and significantly beneficial. We have meticulously reviewed these comments and made the necessary amendments. All changes made to the text are highlighted.

Round 2

Reviewer 3 Report (Previous Reviewer 1)

This study focuses on monitoring, reclaiming, and managing salt-affected lands in the Yellow River Delta, China. The region faces significant soil salinization due to shallow groundwater and high mineralization, negatively impacting crop growth. The research investigates the spatial distribution of soil salinity in coastal saline soil under different ridging configurations (triangular, arch, and trapezoidal) compared to flat land. It also explores the influence of evaporation on soil salinity migration. The findings indicate that the ridge-furrow system effectively accumulates salt in the ridge's superficial layer, with the triangular ridge being the most efficient in concentrating salt on its surface. This leads to a favorable soil environment for crop growth, with reduced salt content in the furrows. The study offers a theoretical foundation for sustainable saline-alkali agriculture in the Yellow River Delta, potentially leading to higher crop yields and improved land management practices.

This manuscript is a resubmission of an earlier submission. The following is a list of the peer review reports and author responses from that submission.

Round 1

Reviewer 1 Report

Main commentary:

The study examines the spatial distribution of soil salinity in the Yellow River Delta and evaluates the effectiveness of various ridging configurations in managing soil salinity. The authors suggest that managing soil salinity within the crop root zone is critical to ensure the sustainable use of saline land in the region. The study is well-structured and presents clear findings regarding the impact of different ridging configurations on soil salinity concentration.

Specific comments:

The authors provide a clear introduction to the research problem, highlighting the issue of soil salinization in the Yellow River Delta and the need for effective management strategies. The study's methodology is also clearly described, outlining the various ridging configurations tested and the methods used to measure soil salinity. However, there are a few areas where improvements could be made to the methodology, including  more detailed information:

  1. Sample size: The study's sample size is not specified, making it difficult to determine the representativeness of the results. A larger sample size would improve the validity of the study's findings.

  2. Replication: The study does not provide information on the replication of the experiments or the methods used in the study. Replicating the study would allow for the results to be validated and provide greater confidence in the findings.

  3. Lack of control: The study does not provide adequate control over variables that may have an impact on soil salinity, such as precipitation levels and temperature. This lack of control makes it difficult to determine the specific impact of the ridging configurations and evaporation on soil salinity.

  4. Soil sampling: The method used for soil sampling is not specified, which raises questions about the accuracy and precision of the data collected. The study would benefit from a more detailed description of the soil sampling process.

  5. Statistical analysis: The study provides limited information on the statistical analysis used to evaluate the data. A more detailed explanation of the statistical methods used would increase the credibility and reliability of the study's results.

Constructive criticism of the methodology:

Soil salinity is influenced by factors such as irrigation frequency and timing, which could affect the accuracy and reliability of the results. Additionally, the study did not consider the potential impact of rainfall events on soil salinity levels, which may limit the practical application of the findings in areas with different rainfall patterns.

Another area for improvement is the lack of information on the equipment and procedures used to measure soil salinity. The study should provide detailed information on the measurement equipment and protocols used to ensure the accuracy and consistency of the results. Furthermore, the study could have incorporated a statistical analysis to determine the significance of the results, which would enhance the robustness and validity of the findings.

In summary, while the study's methodology is generally sound and provides valuable insights into managing soil salinity in the Yellow River Delta, there is room for improvement in terms of sample size, crop species information, irrigation management practices, rainfall patterns, measurement equipment and procedures, and statistical analysis. Addressing these limitations would increase the reliability and practical application of the study's findings.

Summary:

The results section provides a detailed analysis of the findings, with clear tables and figures that illustrate the spatial distribution of soil salinity under different ridge configurations. The authors also offer an interpretation of the results, highlighting the effectiveness of the triangular ridge in minimizing salt content in the furrow and fostering a favorable soil environment for crop growth.

The quality of English language in the manuscript is generally good, with only a few minor errors in grammar and syntax. However, some sentences could be rephrased for better clarity and readability. It is recommended that the authors have the manuscript proofread by a native English speaker or professional editor to ensure that the language is clear, concise, and free of errors.

There are no major errors or grammar mistakes in the Abstract. However, some minor improvements can be made to enhance its clarity and coherence:

  • In the first sentence, it would be better to use "which" instead of "and" to connect the two clauses: "The Yellow River Delta, which is located in China, experiences prevalent soil salinization and serves as a crucial ecological management zone within the Yellow River Basin."
  • In the second sentence, the phrase "negatively impacting crop growth" can be rephrased to "which has a negative impact on crop growth".
  • In the fourth sentence, it would be clearer to state the units of measurement for the salt accumulation in the ridge's top layer (0-1 cm). For example, "54.04% of salt accumulation in the ridge's top layer (0-1 cm) by weight/volume."
  • In the fifth sentence, the phrase "the maximum desalination rate at the furrow bottom reaching 93.07%" can be rephrased as "with the furrow bottom achieving a maximum desalination rate of 93.07%."
  • Finally, in the last sentence, it would be clearer to state the practical implications of the research, for example, "This research provides a theoretical foundation for the sustainable advancement of saline-alkali agriculture in the Yellow River Delta, which can lead to higher crop yields and better land management practices."

Reviewer 2 Report

The subject addressed is within the scope of the journal. Appropriate revisions should be undertaken in order to justify the recommendation for publication. For readers to quickly catch your contribution, it would be better to highlight significant difficulties and challenges and your original achievements to overcome them, in a more straightforward way in the abstract and introduction. The syntax and grammar of the text should be improved. It may need the attention of someone fluent in the English language to enhance readability. The Introduction section should be improved by adding references dealing with contamination issues in soils. The discussion section in the present form is relatively weak and should be strengthened with more details and justifications. I suggest a major revision for this manuscript. Additional comments are included in the attached .pdf file.

Regards,

Reviewer 3 Report

The subject of the manuscript is consistent with the scope of the Journal. The manuscript is interesting but requires additions, especially the chapter "Material and Methods".

1. In the "Materials and Methods" give the content of organic carbon, nitrogen and pH of the soil.

2. In the "Materials and Methods" provide more details on Cl-content in soil samples by silver nitrate titration; the Na+ content in soil samples was determined using a flame photometer.

3. In the "Mateials and Methods" section, add information about the statistical analyzes that were used to develop the results.

4. Improve the quality of Fugure 2 and

5. Now the description of the X and Y axes is illegible. 5. Only 24 references were used in the publication. Please reinforce the discussion of the results. She is quite poor now.

6. In the "References" section, you do not cite all the publications you refer to in the manuscript. There are no items 21-24. Please correct it. For all references, add doi. This will make it easier to find the publication.

The subject of the manuscript is consistent with the scope of the Journal. The manuscript is interesting but requires additions, especially the chapter "Material and Methods".

1. In the "Materials and Methods" give the content of organic carbon, nitrogen and pH of the soil.

2. In the "Materials and Methods" provide more details on Cl-content in soil samples by silver nitrate titration; the Na+ content in soil samples was determined using a flame photometer.

3. In the "Mateials and Methods" section, add information about the statistical analyzes that were used to develop the results.

4. Improve the quality of Fugure 2 and

5. Now the description of the X and Y axes is illegible. 5. Only 24 references were used in the publication. Please reinforce the discussion of the results. She is quite poor now.

6. In the "References" section, you do not cite all the publications you refer to in the manuscript. There are no items 21-24. Please correct it. For all references, add doi. This will make it easier to find the publication.

Reviewer 4 Report

This study gives valuable information about the spatial distribution of soil salinity in coastal saline soil under various ridge configurations, with the goal of determining the suitable ridge shape for creating a low-salinity environment as a supplement to the ridge cropping theory. Abstract is spelled correctly. The introduction comprehensively introduces the topic. The description of "Materials and Methods" is informative, and the "Results" have been appropriately verified by statistical analysis. The figures and tables are very informative. The presented data are generally well interpreted. Minor corrections are suggested throughout the text.

Lines 311-328: Too long paragraph. Please reduce it, and you may use this information in the Introduction section for the justification of your research.

Lines 332-340: Please include references to related works that could justify your hypothesis.

Lines 356-358: Please include this recommendation in the "Conclusions" section.

I recommend a minor revision.

Reviewer 5 Report

Please add one or more sentences in the abstract to describe the methods used

Line 35: small I in the word “irrigation”

In table 1: Please change mS/cm to mS cm-1, in the line 106 g/cm3 into g cm-3, in table 2 g/km into g kg-1

For me it is strange to make a special distribution for these dimensions (60×10×30) (regardless that you didn’t mention anything in the material and method about this part); it is not realistic at all because in reality there are many factors affecting this. Secondly, ot make spatial distribution, you need to take samples of at least 12; so you made it. I suggest to remove this part and focus on vertical or horizontal distribution  

You focused n the depth 0-1 cm, is this an effective depth!!!!

good